# Performance of Quantum Heat Engines Enhanced by Adiabatic Deformation of Trapping Potential

**DOI:** 10.3390/e25030484

**Published:** 2023-03-10

**Authors:** Yang Xiao, Kai Li, Jizhou He, Jianhui Wang

**Affiliations:** 1Department of Physics, Nanchang University, Nanchang 330031, China; 2State Key Laboratory of Surface Physics, Department of Physics, Fudan University, Shanghai 200433, China

**Keywords:** quantum Otto engine, adiabatic deformation, power, efficiency, fluctuations

## Abstract

We present a quantum Otto engine model alternatively driven by a hot and a cold heat reservoir and consisting of two isochoric and two adiabatic strokes, where the adiabatic expansion or compression is realized by adiabatically changing the shape of the potential. Here, we show that such an adiabatic deformation may alter operation mode and enhance machine performance by increasing output work and efficiency, even with the advantage of decreasing work fluctuations. If the heat engine in the sudden limit operates under maximal power by optimizing the control parameter, the efficiency shows certain universal behavior, $\eta^{*}=\eta_{C}/2+\eta_{C}^{2}/8+O\left(\eta_{C}^{3}\right)$, where $\eta_{C}=1-\beta_{h}^{r}/\beta_{c}^{r}$ is the Carnot efficiency, with $\beta_{h}^{r}(\beta_{c}^{r}$) being the inverse temperature of the hot (cold) reservoir. However, such efficiency under maximal power can be produced by our machine model in the regimes where the machine without adiabatic deformation can only operate as a heater or a refrigerator.

## 1. Introduction

Heat engines should ideally have good performance in finite time [1,2,3,4,5,6], and operate stably [7,8,9,10] by exhibiting small fluctuations. Quantum heat engines [11,12,13,14,15,16,17,18,19,20,21,22,23,24,25,26,27,28,29] were observed to operate with novel performance beyond their classical counterparts. These devices with a limited number of freedoms are exposed to not only thermal fluctuations, but also quantum fluctuations related to discrete energy spectra [30,31,32,33,34,35,36]. Both fluctuation mechanisms question the stable operation quantum heat engines [30,32,33]. Thermal design and optimization of quantum heat engines [37,38,39] are, therefore, expected to be considered in order for both good performance and stability, and they constitute one of the central issues in quantum thermodynamics [40,41,42].

To describe the machine performance, there are usually two benchmark parameters: [2,8,17,27,32]: the thermodynamic efficiency $\eta =\langle w\rangle /\langle q_{h}\rangle$, where 〈w〉 is the average work output per cycle and $\langle q_{h}\rangle$ is the average heat released from the hot reservoir, and the power $P=\langle w\rangle /\tau_{cyc}$, with the cycle period $\tau_{cyc}$. Ideally, both these two quantities should have large values for excellent performance, but there is always a power–efficiency trade-off dilemma [43,44,45,46,47,48,49]. An important issue is, hence, that of optimizing the heat engines by ensuring their efficiency under maximal power [1,2,7,11,20,50,51].

Discreteness of energy levels, due to quantization, may significantly improve the performance of a quasi-static quantum Otto cycle [19,31,32,52,53,54] when an inhomogeneous shift of energy levels occurs along an isentropic, adiabatic stroke [54,55]. However, the question as to how such a shift (due to adiabatic deformation of potential) affects a quantum heat engine in the finite-time cycle period, as hinted at in [54], has not previously been answered. Moreover, the random transitions between discrete energy levels are responsible for quantum fluctuations, which dominate at low enough temperatures. A question naturally arises: what is the influence of such adiabatic deformation of potential, related to discrete energy spectra, on the relative power fluctuations that measure engine stability? As we demonstrate, machine efficiency can be improved via controlling the shape of the potential, without sacrificing of machine stability.

In this paper, we study a quantum version of the Otto engine model, which consists of a single particle confined in two different potentials and which works between two heat reservoirs of constant inverse temperatures $\beta_{h}^{r}$ and $\beta_{c}^{r}$ (>$\beta_{h}^{r})$. We analyze the engine performance by determining the efficiency and power with respect to the times spent on the two isochoric strokes and adiabatic deformation parameters. Assuming that only the two lowest energy levels are populated, we show that the adiabatic shape deformation of the potential operates as a heat engine in regions where its counterpart, without the deformation, works as a heater or a refrigerator. We find that the adiabatic deformation enhances performance, as well as stability, through appropriately selecting the forms of the two trapping potentials. We highlight that, in the sudden limit where the total time spent on the two adiabatic strokes is negligible, the efficiency at maximum power of our model shows universal behavior: $\eta^{*}=\eta_{C}/2+\eta_{C}^{2}/8+O\left(\eta_{C}^{3}\right)$, with Carnot efficiency of $\eta_{C}=1-\beta_{h}^{r}/\beta_{c}^{r}$. Yet, such optimized efficiency can be obtained in the regions where the machine, in the absence of adiabatic deformation, cannot operate as a heat engine.

## 2. A Single Particle in a Power-Law Trap

We consider a single particle with mass *m* confined in a one-dimensional power-law trapping potential V(x) along *x* direction with
(1)$$V\left(x\right)∼x^{3/\theta }.$$
This simple class of traps covers, for instance, harmonic (θ=3/2), spherical–quadrupole (θ=3), and infinite potential (θ=0) traps. The Hamiltonian system, $\hat{H}={\hat{p}}^{2}/2m+V\left(x\right)$, with momentum operator $\hat{p}$, can be written in terms of a single particle energy spectrum $\varepsilon_{n}$,
(2)$$\hat{H}=\sum_{n}\varepsilon_{n}{\hat{a}}_{n}^{†}{\hat{a}}_{n},$$
where ${\hat{a}}_{n}^{†}$ (${\hat{a}}_{n}$) is the creation (annihilation) operator, with single-particle quantum number *n*. Thus, ${\hat{a}}_{n}^{†}{\hat{a}}_{n}$ is the particle number operator with quantum number *n*. Throughout the paper we set ℏ≡1 for simplicity. The energy spectrum can be written as $\varepsilon_{n}=\langle n|\hat{H}|n\rangle =\omega n^{\sigma }$. Here, ω is the energy gap between the ground state and the first excited state (which we call the energy gap, for simplicity, in what follows), and σ (>0) is called the potential exponent [56], which is determined by the parameter θ in Equation (Equation 1) (For the one-dimensional trapped system in the *x* direction, the time-independent Schrödinger equation may be written as $\left[{\hat{p}}^{2}/2m+V\left(x\right)\right]\Psi_{n}\left(x\right)=\varepsilon_{n}\Psi_{n}\left(x\right)$, where $\varepsilon_{n}$ are the energy eigenvalues and $\Psi_{n}$ are eigenfunctions. For example, for a single particle confined in a box trap which reads V(x)=0 for 0≤x≤L and V(x)=∞ [due to θ=0 with θ defined in Equation (Equation 1)] otherwise, the energy spectrum is obtained as $\varepsilon_{n}=n^{2}\pi^{2}/\left(2mL^{2}\right)$. When the single particle is confined in a harmonic trap with θ=2/3, the potential becomes $V\left(x\right)=\omega x^{2}/2$, leading to $\varepsilon_{n}=\left(1/2+n\right)\omega$. The energy eigenvalues $\varepsilon_{n}=\omega n^{\sigma }$ are determined by the trapping potential V(x), and the shape of trapping potential associated with θ can be captured well by the so-called trap exponent σ.) and is, thus, dependent on the shape of the external potential. For example, for a one-dimensional harmonic trap σ=1 and $\varepsilon_{n}=n\omega$, where ω is the trap frequency, and for a one-dimensional infinite deep potential (also called a one-dimensional box trap) with length *L*, σ=2 and $\varepsilon_{n}=n^{2}\omega$, where $\omega \equiv \pi^{2}/\left(2mL^{2}\right)$.

The expressions for creation and annihilation operators (${\hat{a}}_{n}^{†}$ and ${\hat{a}}_{n}$) in Equation (Equation 2) depend on the trapping potential under consideration. A typical example is that a single particle is confined in a one-dimensional infinite potential well, which is given by V(x)=0 for 0≤x≤L and V(x)=∞ otherwise. The state wave function of a trapped particle reads $\Psi_{n}\left(s\right)=|n\rangle =\sqrt{2/L}\;\sin\left(ns\right)$ with s=πx/L when 0≤s≤π. The creation and annihilation operators for the system should satisfy ${\hat{a}}_{n}^{†}|n\rangle ∼|n+1\rangle$ and ${\hat{a}}_{n}|n\rangle ∼|n-1\rangle$ [57]. In view of the fact that $d\psi_{n}\left(s\right)/ds=\sqrt{2/L}\;n\cos\left(ns\right)$, we define the creation and annihilation operators as ${\hat{a}}_{n}=\sqrt{\hat{N}}\cos\left(s\right)-\sqrt{{\hat{N}}^{-1}}\sin\left(s\right)\frac{d}{ds}$ and ${\hat{a}}_{n}^{†}=\sqrt{\hat{N}+1}\cos\left(s\right)+\sqrt{\hat{N}+1}{\hat{N}}^{-1}\sin\left(s\right)\frac{d}{ds}$. Here, $\hat{N}$ is the number operator defined by $\hat{N}|n\rangle =\sqrt{n}|n\rangle$ and its inverse ${\hat{N}}^{-1}|n\rangle =n^{-1}|n\rangle$. By using these definitions, we obtain ${\hat{a}}_{n}|n\rangle =\sqrt{n}|n-1\rangle$ and ${\hat{a}}_{n}^{†}|n\rangle =\sqrt{n+1}|n+1\rangle$. We then find that the commutator is $[{\hat{a}}_{n},{\hat{a}}_{n}^{†}]=1$, and the system energy becomes $\langle \hat{H}\rangle =\sum_{n}n^{2}/\left(2mL^{2}\right)\langle {\hat{a}}_{n}^{†}{\hat{a}}_{n}\rangle$, where $\langle {\hat{a}}_{n}^{†}{\hat{a}}_{n}\rangle$ corresponds to the occupation probability at state *n* for the single-particle system.

The state of the system at thermal equilibrium with a heat bath of inverse temperature β can be described by the canonical form $\hat{\rho }=\sum_{n}p_{n}\left|n\rangle \langle n\right|=Z^{-1}\exp\left(-\beta \hat{H}\right)$, where $p_{n}=e^{-\beta \varepsilon_{n}}/Z$ is the probability of finding the system in state |n〉, with the partition function $Z=\mathrm{Tr}(e^{-\beta \hat{H}})$. The system entropy reads $S=-\mathrm{Tr}(\hat{\rho }\ln\hat{\rho })$, where $\hat{\rho }=\hat{\rho }\left(\beta \omega ,\sigma \right)$, and, thus, the entropy takes the form of S=S(βω,σ). For the gas in a given trap, the entropy *S* merely depends on the parameter βω: S=S(βω), and in an adiabatic process βω= constant. However, an adiabatic deformation of trap, by changing σ, leads to change in the parameter ‘βω’ [54,55] to keep entropy *S* constant. That is, a quantum adiabatic process where the entropy is kept constant can be realized via changing the shape of the trapping potential.

## 3. General Expressions of Efficiency and Power for Quantum Otto Engines with Deformation of Trapping Potential

In contrast to conventional quantum heat engines, where the working substance is confined in a given form of trap, the quantum engine under consideration works, based on two different forms of one-dimensional trapping potentials V(x), by adopting two different values of θ in Equation (Equation 1). The quantum Otto engine, sketched in Figure 1, consists of four consecutive strokes, as outlined in the following: (i) Hot isochoric stroke A→B. The single particle is confined in a one-dimensional trap along the *x* direction with $\theta =\theta_{h}$, and the trapped system is weakly coupled to a hot reservoir of constant inverse temperature $\beta_{h}^{r}$ in time duration $\tau_{h}$. Since the external field V(x) is frozen, the energy gap is kept constant at $\omega =\omega_{h}$; (ii) Adiabatic expansion B→C. The von Neumann entropy of the system is constant along the adiabatic stroke in which the system evolution is unitary. While the system is isolated from the heat reservoir in time $\tau_{hc}$, and the form of the potential gradually changes from the trap $V\left(x\right)∼x^{3/\theta_{h}}$ to the trap $V\left(x\right)∼x^{3/\theta_{c}}$ by tuning θ; (iii) Cold isochoric stroke C→D. Both the trap configuration and the energy gap are kept fixed, namely, $\omega =\omega_{c}$ and $\theta =\theta_{c}$. Within a time interval of $\tau_{c}$, the trapped system is weakly coupled to a cold reservoir with constant inverse temperature $\beta_{c}^{r}\left(>\beta_{h}^{r}\right)$; (iv) Adiabatic compression D→A. The system is again isolated from the heat reservoir in time duration $\tau_{ch}$, and the trap configuration changes gradually from the trap $V\left(x\right)∼x^{3/\theta_{c}}$ to the trap $V\left(x\right)∼x^{3/\theta_{h}}$. During the hot or cold isochoric strokes, the system would relax to the thermal state at the ending instant B(D) of the hot (cold) isochore, if $\tau_{h}\left(\tau_{c}\right)$ is long enough. The times allocated to the four strokes set the total cycle period, $\tau_{cyc}=\tau_{h}+\tau_{c}+\tau_{hc}+\tau_{ch}$.

For the Otto cycle, the work is produced only in the two adiabatic branches, with heat produced alongside the isochoric processes. Initially, the time is assumed to be t=0. The Hamiltonian system changes from $\hat{H}\left(\tau_{h}\right)$ to $\hat{H}\left(\tau_{h}+\tau_{hc}\right)$ along the adiabatic expansion B→C, and it goes back to $\hat{H}\left(0\right)$ from $\hat{H}\left(\tau_{cyc}-\tau_{ch}\right)$ after the adiabatic compression D→A. The Hamiltonian system is kept constant along each isochoric stroke, namely, $\hat{H}\left(0\right)=\hat{H}\left(\tau_{h}\right)$ and $\hat{H}\left(\tau_{h}+\tau_{hc}\right)=\hat{H}\left(\tau_{cyc}-\tau_{ch}\right)$. The stochastic work done by the system, per cycle, is, thus, the total work output along the two adiabatic trajectories [32,58], which reads $w[\hat{H}\left(\tau_{h}\right)|n\rangle ;\hat{H}\left(\tau_{cyc}-\tau_{ch}\right)\left|m\rangle \right]=\left[\langle n\right|\hat{H}\left(\tau_{h}\right)|n\rangle -\langle n|\hat{H}\left(\tau_{h}+\tau_{hc}\right)\left|n\rangle \right]+\left[\langle m\right|\hat{H}\left(\tau_{cyc}-\tau_{ch}\right)|m\rangle -\langle m|\hat{H}\left(0\right)\left|m\rangle \right]$. The stochastic work for the engine cycle is then given by
(3)$$w[|n\left(\tau_{h}\right)\rangle ;|m\left(\tau_{cyc}-\tau_{ch}\right)\rangle ]=\varepsilon_{n}^{h}-\varepsilon_{n}^{c}+\varepsilon_{m}^{c}-\varepsilon_{m}^{h}.$$
where we used $\varepsilon_{i}^{h}=\langle i|\hat{H}\left(0\right)|i\rangle =\langle i|\hat{H}\left(\tau_{h}\right)|i\rangle$, $\varepsilon_{i}^{c}=\langle i|\hat{H}\left(\tau_{h}+\tau_{hc}\right)|i\rangle =\langle i|\hat{H}\left(\tau_{cyc}-\tau_{ch}\right)|i\rangle$, with i=m,n.

During the adiabatic stroke, the level populations do not change, $p_{n,B}=p_{n,C}$ and $p_{m,A}=p_{m,D}$, and the probability density of the stochastic work *w* can then be determined according to
(4)$$p\left(w\right)=\sum_{n,m}p_{n,B}p_{m,A}\delta \{w-w[|n\left(\tau_{h}\right)\rangle ;|m\left(\tau_{cyc}-\tau_{ch}\right)\rangle ]\},$$
where δ(·) is the Dirac’s δ function. The average work output per cycle, 〈w〉=∫wp(w)dw, can be obtained as
(5)$$\langle w\rangle =\langle {\hat{H}}_{B}\rangle -\langle {\hat{H}}_{C}\rangle +\langle {\hat{H}}_{D}\rangle -\langle {\hat{H}}_{A}\rangle .$$
Here, and hereafter, we use the subscripts A,B,C, and *D* (in Figure 1) to indicate the physical quantity at times $t=0,\tau_{h},\tau_{h}+\tau_{hc},$ and $t=\tau_{cyc}-\tau_{c}$, respectively. We define the dimensionless energy *g* as $g\equiv \mathrm{Tr}\left(\hat{\rho }\hat{H}\right)/\left(\omega \right)$, which, according to Equation (Equation 2), can be expressed as
(6)$$g=g\left(\beta \omega ,\sigma \right)=\sum_{n}n^{\sigma }\langle {\hat{a}}_{n}^{†}{\hat{a}}_{n}\rangle .$$
As emphasized, ${\hat{a}}_{n}^{†}{\hat{a}}_{n}$ indicates the occupation number operator of a given state *n*, and, thus, $\langle {\hat{a}}_{n}^{†}{\hat{a}}_{n}\rangle$ is the average occupation number at state *n*. While the trapping potentials are $V\left(x\right)∼x^{3/\theta_{h}}$ and $V\left(x\right)∼x^{3/\theta_{c}}$ in the hot and cold isochoric strokes, respectively, we use the trap exponents $\sigma_{h}$ and $\sigma_{c}$ (rather than $\theta_{h}$ and $\theta_{c}$) to characterize the forms of the trapping potential in what follows.

To describe the degree of the shape deformation of the trapping potential, we introduce the deformation parameters for adiabatic compression and expansion which are defined by
(7)$$\xi_{hc}\equiv \frac{g_{C}}{g_{B}}=\frac{g(\sigma_{c},\beta_{C}\omega_{c})}{g(\sigma_{h},\beta_{B}\omega_{h})},\xi_{ch}\equiv \frac{g_{A}}{g_{D}}=\frac{g(\sigma_{h},\beta_{A}\omega_{h})}{g(\sigma_{c},\beta_{D}\omega_{c})},$$
respectively. Except in the special case when the shape of the potential is not changed along the engine cycle with $\sigma_{c}=\sigma_{h}$ and $\xi_{hc}=\xi_{ch}=1$, these parameters, $\xi_{hc}$ and $\xi_{ch}$, depend on the trap exponents $\sigma_{c}$ and $\sigma_{h}$, and, thus, they capture all information about the adiabatic deformation of trapping potential. We also note that, in the presence of adiabatic deformation, the deformation parameters, $\xi_{ch}$ and $\xi_{hc}$, would be affected by the times, $\tau_{c}$ and $\tau_{h}$, since the so-called system temperatures, at instants A,B,C, and *D* in Figure 1, are dependent on these times $\tau_{c}$ and $\tau_{h}$.

Using Equations (Equation 5)–(Equation 7), we find that the average work takes the form of
(8)$$\langle w\rangle =\left(\omega_{h}-\omega_{c}\frac{\xi_{hc}g_{B}-g_{D}}{g_{B}-\xi_{ch}g_{D}}\right)\left(g_{B}-\xi_{ch}g_{D}\right).$$
The work fluctuations can be determined according to
(9)$$\langle \delta^{2}w\rangle =\langle w^{2}\rangle -{\langle w\rangle }^{2},$$
where $\langle w^{2}\rangle =\int w^{2}p\left(w\right)dw=\sum_{n,m}p_{n,B}p_{m,A}{\left(\varepsilon_{n}^{h}-\varepsilon_{n}^{c}+\varepsilon_{m}^{c}-\varepsilon_{m}^{h}\right)}^{2}$.

Based on the two-time measurement approach, the probability density function of the stochastic heat $q_{h}$ along the hot isochoric stroke, where no work is produced, can be determined by the conditional probability to arrive at
(10)$$p\left(q_{h}\right)=\sum_{n,m}p_{m\to n}^{\tau_{h}}p_{m,A}\delta \left[q_{h}-\left(\varepsilon_{n}^{h}-\varepsilon_{m}^{h}\right)\right],$$
where $p_{m,A}$ is the probability that the system is initially in state *m* at time t=0, and $p_{m\to n}^{\tau_{h}}$ is the probability of the system collapsing into another state *n* after a time period $\tau_{h}$. Here $p_{m\to n}^{\tau_{h}}{|}_{\tau_{h}\to \infty }=p_{n}^{eq}\left(\beta_{h}^{r}\right)$, where $p_{n}^{eq}\left(\beta_{h}^{r}\right)=e^{-\beta_{h}^{r}\varepsilon_{n}^{h}}/\left(e^{-\beta_{h}^{r}\varepsilon_{m}^{h}}+e^{-\beta_{h}^{r}\varepsilon_{n}^{h}}\right)$. For each cycle, heat is transferred only in the isochore, while work is produced only along the adiabatic process. The heat absorbed from the hot bath is given by $\langle q_{h}\rangle =\langle {\hat{H}}_{B}\rangle -\langle {\hat{H}}_{A}\rangle$, or
(11)$$\langle q_{h}\rangle =\omega_{h}\left(g_{B}-\xi_{ch}g_{D}\right).$$
Due to energy conservation, the heat discharged to the cold reservoir along the cold isochoric stroke can be directly calculated according to $\langle q_{c}\rangle =\langle q_{h}\rangle -\langle w\rangle$ (see also Figure 1).

In order to evaluate the average values of heat and work in a finite time cycle, we should analyze the system dynamics along two isochoric strokes to derive average work and heat. We use $\Gamma_{c}\left(\Gamma_{h}\right)$ to denote the thermal conductivity between the system and cold (hot) heat reservoir and introduce $x=e^{-\Gamma_{h}\tau_{h}}$ and $y=e^{-\Gamma_{c}\tau_{c}}$. We show that these quantities, (Equation 8) and (Equation 11), can be expressed as a function of *x* and *y* (see Appendix A for details),
(12)$$\begin{array}{ccc} \langle w\rangle & = & \left(\frac{1-\xi_{hc}\xi_{ch}y}{1-y}\omega_{h}-\xi_{hc}\omega_{c}\right)\\ & \times & \left(g_{h}^{eq}-\frac{\xi_{ch}\omega_{h}-\frac{1-\xi_{hc}\xi_{ch}x}{1-x}\omega_{c}}{\frac{1-\xi_{hc}\xi_{ch}y}{1-y}\omega_{h}-\xi_{hc}\omega_{c}}g_{c}^{eq}\right)G, \end{array}$$
and
(13)$$\langle q_{h}\rangle =\omega_{h}\left(\frac{1-\xi_{hc}\xi_{ch}y}{1-y}g_{h}^{eq}-\xi_{ch}g_{c}^{eq}\right)G,$$
where we used $G=\frac{\left(1-x)(1-y\right)}{\left(1-\xi_{hc}\xi_{ch}xy\right)}$. The heat quantity released into the cold bath can be directly calculated by $\langle q_{c}\rangle =\langle w\rangle -\langle q_{h}\rangle$, due to the conservation of energy. In the absence of adiabatic deformation, these average values, (Equation 12) and (Equation 13), reduce to $\langle w\rangle =\left(\omega_{h}-\omega_{c}\right)\left(g_{h}^{eq}-g_{c}^{eq}\right)G,$ and $\langle q_{h}\rangle =\omega_{h}\left(g_{h}^{eq}-g_{c}^{eq}\right)G$. In such a case, we present these formulae in a broader context by considering a power-law trap in which *g* may not be the mean population if $\sigma_{c,h}\neq 1$. We reproduce the result obtained from the harmonic trap, where $\sigma_{c}=\sigma_{h}=1$, and, thus, *g* denotes the average population.

The thermodynamic efficiency$,\eta =\langle w\rangle /\langle q_{h}\rangle$, then follows as
(14)$$\eta =1-\frac{\omega_{c}}{\omega_{h}}\frac{\xi_{hc}g_{h}^{eq}-\frac{1-\xi_{hc}\xi_{ch}x}{1-x}g_{c}^{eq}}{\frac{1-\xi_{hc}\xi_{ch}y}{1-y}g_{h}^{eq}-\xi_{ch}g_{c}^{eq}},$$
which simplifies to $\eta =1-\frac{\omega_{c}}{\omega_{h}}\frac{\xi_{hc}g_{h}^{eq}-g_{c}^{eq}}{g_{h}^{eq}-\xi_{ch}g_{c}^{eq}}$ in the quasi-static limit where $\tau_{c}\to \infty$ and $\tau_{h}\to \infty$. In the case when the shape of the potential is adiabatically changed, an inhomogeneous shift of energy levels is created, resulting in thermodynamic efficiency (Equation 14) that depends on the shapes of the potentials along two isochoric strokes, excepting the case when the two potentials are identical to each other, which would result in efficiency reducing to that of the cycles without adiabatic shape deformation, $\eta =1-\omega_{c}/\omega_{h}$.

## 4. Performance and Stability of a Two-Level Machine

The efficiency may be enhanced by adiabatically changing the form of the potential. To better understand the influence induced by adiabatic deformation on the performance of thermal machine, we investigate how the adiabatic deformation affects the efficiency and the power. In this section, we consider, as an example, the Otto engine working in the low-temperature limit, by assuming that only the two lowest energy levels are appreciably populated. We show in Appendix B that, for the two-level engine where the system Hamiltonian (Equation 2) simplifies to $\hat{H}=\omega \left({\hat{a}}_{1}^{†}{\hat{a}}_{1}+2^{\sigma }{\hat{a}}_{2}^{†}{\hat{a}}_{2}\right)$ and the system energy becomes $\langle \hat{H}\rangle =\omega \sum_{n=1,2}n^{\sigma }\langle {\hat{a}}_{n}^{†}{\hat{a}}_{n}\rangle$, the deformation parameters $\xi_{ch}$ and $\xi_{hc}$ defined by Equation (Equation 7) take the forms of
(15)$$\begin{array}{ccc} \xi_{hc} & = & \frac{\gamma_{c}-1}{\gamma_{h}-1}+\frac{\gamma_{h}-\gamma_{c}}{\gamma_{h}-1}\\ & \times & \frac{1}{\left[g_{h}^{eq}+\left(\frac{\gamma_{c}-\gamma_{h}}{\gamma_{c}-1}+\frac{\gamma_{h}-1}{\gamma_{c}-1}g_{c}^{eq}-g_{h}^{eq}\right)\frac{(1-y)x}{1-xy}\right]},\\ \xi_{ch} & = & \frac{\gamma_{h}-1}{\gamma_{c}-1}+\frac{\gamma_{c}-\gamma_{h}}{\gamma_{c}-1}\\ & \times & \frac{1}{\left[g_{c}^{eq}+\left(\frac{\gamma_{h}-\gamma_{c}}{\gamma_{h}-1}+\frac{\gamma_{c}-1}{\gamma_{h}-1}g_{h}^{eq}-g_{c}^{eq}\right)\frac{(1-x)y}{1-xy}\right]}, \end{array}$$
where $g_{c}^{eq}=\frac{e^{-\beta_{c}^{r}\omega_{c}}+\gamma_{c}e^{-\gamma_{c}\beta_{c}^{r}\omega_{c}}}{e^{-\beta_{c}^{r}\omega_{c}}+e^{-\gamma_{c}\beta_{c}^{r}\omega_{c}}}$, $g_{h}^{eq}=\frac{e^{-\beta_{h}^{r}\omega_{h}}+\gamma_{h}e^{-\gamma_{h}\beta_{h}^{r}\omega_{h}}}{e^{-\beta_{h}^{r}\omega_{h}}+e^{-\gamma_{h}\beta_{h}^{r}\omega_{h}}}$, $\gamma_{h}=2^{\sigma_{h}}$, and $\gamma_{c}=2^{\sigma_{c}}$. Substituting Equation (Equation 15) into Equations (Equation 12) and (Equation 14), it follows that the average work (Equation 12) and thermodynamic efficiency (Equation 14) of the two-level machine in finite time are given by
(16)$$\begin{array}{ccc} \langle w\rangle & = & \left(g_{h}^{eq}-\frac{\gamma_{h}-1}{\gamma_{c}-1}g_{c}^{eq}+\frac{\gamma_{h}-\gamma_{c}}{\gamma_{c}-1}\right)\\ & \times & \left(\omega_{h}-\frac{\gamma_{c}-1}{\gamma_{h}-1}\omega_{c}\right)G, \end{array}$$
and
(17)$$\eta =1-\frac{\omega_{c}}{\omega_{h}}\frac{\gamma_{c}-1}{\gamma_{h}-1}.$$
Note that when $\gamma_{c}<\gamma_{h}$, the efficiency (Equation 17) is larger than the efficiency without adiabatic deformation ($\eta =1-\omega_{c}/\omega_{h})$. For the two-level engine, the work fluctuation $\langle \delta^{2}w\rangle$ (Equation 9) can be analytically obtained as
(18)$$\begin{array}{ccc} \langle \delta^{2}w\rangle & = & \langle w^{2}\rangle -{\langle w\rangle }^{2}\\ & = & \frac{\gamma_{h}-1}{\gamma_{c}-1}{\left[\omega_{h}-\omega_{c}\frac{\gamma_{c}-1}{\gamma_{h}-1}\right]}^{2}\\ & \times & \left[\left(g_{B}-1\right)\left(\gamma_{c}-g_{D}\right)+\left(g_{D}-1\right)\left(\gamma_{h}-g_{B}\right)\right]\\ & - & {\left\{\frac{1}{\gamma_{c}-1}\left[\omega_{h}-\omega_{c}\frac{\gamma_{c}-1}{\gamma_{h}-1}\right]\right\}}^{2}\\ & \times & {\left[\left(g_{B}-1\right)\gamma_{c}-\left(g_{D}-1\right)\gamma_{h}+\left(g_{D}-g_{B}\right)\right]}^{2}, \end{array}$$
where $g_{B}$ and $g_{D}$ are $g_{B}=g_{h}^{eq}+\left(\frac{\gamma_{c}-\gamma_{h}}{\gamma_{c}-1}+\frac{\gamma_{h}-1}{\gamma_{c}-1}g_{c}^{eq}-g_{h}^{eq}\right)\frac{(1-y)x}{1-xy}$ and $g_{D}=g_{c}^{eq}+\left(\frac{\gamma_{h}-\gamma_{c}}{\gamma_{h}-1}+\right.$

$\left.\frac{\gamma_{c}-1}{\gamma_{h}-1}g_{h}^{eq}-g_{c}^{eq}\right)\frac{(1-x)y}{1-xy}.$ When adiabatic deformation is absent, the work fluctuations turn out to be $\langle \delta^{2}w\rangle ={\left(\omega_{h}-\omega_{c}\right)}^{2}\left[\left(g_{B}-1\right)\left(\gamma_{c}-g_{D}\right)+\left(g_{D}-1\right)\left(\gamma_{h}-g_{B}\right)\right]-{\left\{\frac{1}{\gamma_{c}-1}\left[\omega_{h}-\omega_{c}\frac{\gamma_{c}-1}{\gamma_{h}-1}\right]\right\}}^{2}\times {\left[\left(g_{B}-1\right)\gamma_{c}-\left(g_{D}-1\right)\gamma_{h}+\left(g_{D}-g_{B}\right)\right]}^{2},$

The system reaches thermal equilibrium at the end of the hot or cold isochore when the process is in the quasi-static limit. In this case, where x→0, y→0, and $G=\frac{\left(1-x)(1-y\right)}{1-xy}\to 1$, the work (Equation 16) and work fluctuations (Equation 18) turn out to be
(19)$$\begin{array}{ccc} \langle w\rangle & = & \left(g_{h}^{eq}-\frac{\gamma_{h}-1}{\gamma_{c}-1}g_{c}^{eq}+\frac{\gamma_{h}-\gamma_{c}}{\gamma_{c}-1}\right)\\ & \times & \left(\omega_{h}-\frac{\gamma_{c}-1}{\gamma_{h}-1}\omega_{c}\right), \end{array}$$
(20)$$\begin{array}{ccc} \langle \delta^{2}w\rangle & = & \frac{\gamma_{h}-1}{\gamma_{c}-1}{\left(\omega_{h}-\omega_{c}\frac{\gamma_{c}-1}{\gamma_{h}-1}\right)}^{2}\\ & \times & \left[\left(g_{h}^{eq}-1\right)\left(\gamma_{c}-g_{c}^{eq}\right)+\left(g_{c}^{eq}-1\right)\left(\gamma_{h}-g_{h}^{eq}\right)\right]\\ & - & {\left\{\frac{1}{\gamma_{c}-1}\left(\omega_{h}-\omega_{c}\frac{\gamma_{c}-1}{\gamma_{h}-1}\right)\right\}}^{2}\\ & \times & {\left[\left(g_{h}^{eq}-1\right)\gamma_{c}-\left(g_{c}^{eq}-1\right)\gamma_{h}+\left(g_{c}^{eq}-g_{h}^{eq}\right)\right]}^{2}. \end{array}$$
The power output and power fluctuations are then determined according to $P=\langle w\rangle /\tau_{cyc}$ and $\delta^{2}P=\langle \delta^{2}w\rangle /\tau_{cyc}^{2}$.

In Figure 2a we plot the normalized efficiency $\eta /\eta_{C}$ at the quasi-static limit as a function of the ratio *r* (with $r\equiv \sqrt{\omega_{h}/\omega_{c}}$) in the presence of adiabatic shape deformation, comparing the corresponding result for the Otto engine without deformation of trap. In the absence of adiabatic deformation of trap $\left(\gamma_{c}=\gamma_{h}\right)$, the three different conditions of the compression ratio *r* correspond to the three modes of the machine: (1) for r≤1, the machine operates as a heater, (2) for $1<r\leq r_{C}\equiv \sqrt{\beta_{c}^{r}/\beta_{h}^{r}}$, it works as a heat engine, and (3) for $r>r_{C}$, it becomes a refrigerator. However, when adiabatically changing the shape of the trapping potential, the machine can operate as a heat engine even in boundaries (1) and (3). Figure 2b–d show contour plots of the average work 〈w〉 versus $\omega_{h}$ and $\omega_{c}$ for different values of $\gamma_{c,h}$. The color areas indicate the positive work of the thermal machine as a heat engine, showing that the positive work condition changed due to adiabatic deformation of trapping potential.

For complete thermalization along each isochore, Figure 3a displays the average work (Equation 19) and the work fluctuations (Equation 20) as a function of the compression ratio *r*, respectively, for $\gamma_{c}=\gamma_{h}=2,2\gamma_{c}=\gamma_{h}=4$, and $\gamma_{c}=2\gamma_{h}=4$. While the efficiency is improved by increasing *r* for given $\gamma_{c}$ and $\gamma_{h}$, the average work 〈w〉 first increases, and then decreases as the ratio *r* increases. The behavior of curves for work fluctuations $\langle \delta^{2}w\rangle$ as a function of *r* is dependent on $\gamma_{c}$ and $\gamma_{h}$. It can be observed from Figure 3a that, while for $\gamma_{c}\leq \gamma_{h}$ the curve of work fluctuations, as a function of *r*, is linear, it becomes parabolic when $\gamma_{c}>\gamma_{h}$. Both the work fluctuations $\langle \delta^{2}w\rangle$ and average work 〈w〉 for $\gamma_{c}>\gamma_{h}$ are much smaller than those obtained from the case when $\gamma_{c}\leq \gamma_{h}$. Figure 3a also shows that the regime of positive work (〈w〉>0) is sensitively dependent on the values of the parameters $\gamma_{c}$ and $\gamma_{h}$, and the presence of adiabatic deformation changes the positive work condition for the quantum engine.

The coefficient of variation for power $f_{P}=\sqrt{\delta^{2}P}/P$, equivalent to the square root of the relative work fluctuations,$\sqrt{\langle \delta^{2}w\rangle }/\langle w\rangle$, is also called the relative power fluctuation. This coefficient measures the dispersion of the probability distribution and, thus, can describe the machine stability [17]. Comparing the efficiency and relative power fluctuations of the engine with $\gamma_{c}=2$ ($\gamma_{c}=4$) to each other, Figure 3b shows that optimization of the quantum heat engine can be realized by selecting the appropriate form of trapping potential during the hot isochoric stroke. For example, the engine with $\gamma_{c}=\gamma_{h}=4$ works at efficiency η=0.76 and relative power fluctuation $f_{P}=25.6$, but the model with $\gamma_{c}=4$ and $\gamma_{h}=3.6$ operates under η=0.73 and $f_{P}=12.7$. That is, when $\gamma_{c}=4$, the relative power fluctuation $f_{p}$ for $\gamma_{h}=3.6$ is halved as compared to its value in the absence of adiabatic deformation, while efficiency only slightly decreased. Another example to consider $\gamma_{c}=2$ is comparing $\gamma_{h}=2$ with $\gamma_{h}=1.84$. In contrast to the former case, where η=0.76 and $f_{P}=13.77$, in the latter case η=0.72 and $f_{P}=6.9$, showing again that the relative power fluctuations can be significantly decreased with a particularly small decrease in efficiency. More importantly, by suitably choosing the shapes of the trapping potential, we may even design an engine model that runs more stably and effectively, as also shown in Figure 3b. A typical example is that of the engine of $\gamma_{c}=4$ and $\gamma_{h}=3.35$, producing efficiency η=0.7 with $f_{P}=9.15$, but the model with $\gamma_{c}=2$ and $\gamma_{h}=1.9$ runs at η=0.74 and $f_{P}=8.53$. By comparison, the latter model shows better overall performance than the former, since it runs more stably by decreasing relative power fluctuations, even with higher efficiency η. Hence, adiabatically changing the form of the trapping potential may even contribute to a decrease in the relative power fluctuations with an increase in efficiency, when compared with an engine without adiabatic deformation. So, for quantum engines, having selected suitable machine parameters, adiabatic deformation along an engine cycle may be an effective optimization approach to significantly improve engine stability.

Both Figure 2 and Figure 3 show that the average work, work fluctuations, coefficient of variation for power, and even positive work condition, are strongly affected by change in the values of $\gamma_{c}$ and $\gamma_{h}$. To further see clearly how the adiabatic deformation quantitatively affects the performance and fluctuations for the engine, we plotted average work, work fluctuations, and coefficient of variation for power as a function of Carnot efficiency in Figure 4a–c, respectively, where the value of $\gamma_{c}$ was kept fixed $\left(\gamma_{c}=2\right)$, and the value of $\gamma_{h}$ was slightly changed ($\gamma_{h}=2,1.8,2.2)$. Figure 4a demonstrates that, in the positive work region, adiabatic deformation could increase the average work in the certain regime of $\eta_{C}$, though it may decrease the work when $\eta_{C}$ was relatively large. The fluctuations of the heat engine, including the work fluctuations and the relative work fluctuations (coefficient of variation for power), were always significantly decreased by the adiabatic deformation [as per Figure 4b,c]. These figures show that the shape change of the trap may enhance the average work, unless the difference between the two bath reservoirs is particularly large, while it always enhances the machine stability captured by the fluctuations. As a specific example, at $\eta_{C}=0.85$, both the work and efficiency [see Equation (Equation 17)] could be enlarged, but the fluctuations of work and relative power decreased. In the present case, the adiabatic deformation was, thus, an essential ingredient in improving the engine performance and stability in a certain regime.

## 5. The Engine under Maximal Power Output

Since the power output, $P=\langle w\rangle /\tau_{cyc}$, would vanish if the cycle was quasi-static and the cycle period approached infinity, the engine should operate, practically, in finite time to produce finite power output. In this section we consider the efficiency and power statistics for the two-level machine under maximum power by optimizing power with respect to external degrees, and assuming that the two adiabatic strokes are realized in the sudden limit [59,60]. It is not difficult to verify that the efficiency at maximum power $\eta^{*}$ can be determined by using the method shown in Appendix B to analytically obtain [11] $\eta_{anal}^{*}=\eta_{C}^{2}/\left[\eta_{C}-\left(1-\eta_{C}\right)\ln\left(1-\eta_{C}\right)\right]=\eta_{C}/2+\eta_{C}^{2}/8+O\left(\eta_{C}^{3}\right)$, which shows the same universality with the CA efficiency [1,3,48] $\eta_{CA}=\eta_{C}/2+\eta_{C}^{2}/8+O\left(\eta_{C}^{3}\right)$.

In Figure 5 we plotted the analytical efficiency of maximum power $\eta_{anal}^{*}$ as a function of the $\eta_{C}$, comparing the exact numerical result for different values of $\gamma_{c,h}$ and the CA efficiency $\eta_{CA}$. These curves of the optimal efficiency for different $\gamma_{c,h}$, together with the analytical expression of $\eta_{anal}^{*}$, collapse into a single line, and they are in nice agreement with the CA efficiency $\eta_{CA}$. It was, therefore, shown that the efficiency at maximum power, agreeing well with $\eta_{CA}$, is independent of the shapes of the two trapping potentials. As emphasized, the heat engine under consideration can proceed at such efficiency in regimes where the machine, without adiabatic deformation, may only operate as a heater or a refrigerator.

## 6. Conclusions

As a result of energy quantization, a quantum adiabatic process can be realized by changing the shape of the trapping potential. Such a shape deformation causes the classical limit, where the principle of the equipartition of energy holds, to vanish and is, therefore, of purely quantum origin. Here, we investigated the performance and fluctuations in quantum Otto engines in the presence of adiabatic deformation of one power-trap potential. We started using stochastic thermodynamics and the quantum master equation to determine heat and work statistics, and then presented general expressions for time-dependent efficiency and work [cf. Equations (Equation 12) and (Equation 14)], in which adiabatic deformation parameters of the two adiabatic strokes are involved.

We proposed an exact analytical description for the performance and fluctuations in these quantum engines at the low-temperature limit where only the lowest two energy levels are occupied. We showed that quantum heat engines, with adiabatic deformation, can run in the extended regimes where their counterparts, without adiabatic deformation, operate as heaters or refrigerators. Examining the efficiency and coefficient of variation of power, we found that an appropriate selection of two trapping potentials enables engines to be built that are capable of performing more stably and efficiently. We also showed that, even for a given trap in an isochore, the relative power fluctuations in our engines are significantly smaller than those of engines in the absence of adiabatic deformation, with higher efficiency than that of engines without adiabatic deformation. By tuning the energy gap between the ground and the first excited state, we found that the efficiency at maximum power is independent of shape deformation and shares the same universality with the CA efficiency. This optimized efficiency, however, can be realized in regions where engines experiencing no change in the shape of the potentials cannot operate as heat engines.

Our approach can be directly used to describe an ensemble of many non-interacting particles (with particle number *N*) confined in a d−dimensional power-law trap. In such a case, the Hamiltonian system becomes $\hat{H}=\sum_{n}\varepsilon_{n}{\hat{a}}_{n}^{†}{\hat{a}}_{n}$,where $\varepsilon_{n}=\langle n|\hat{H}|n\rangle$, with $n=n_{1},\cdots ,n_{d}$ and $N=\sum_{n}{\hat{a}}_{n}^{†}{\hat{a}}_{n}$. Following the approach adopted in this paper, we can reproduce the same results and arrive at the same conclusions. Our results significantly add to the study of quantum heat engines in finite time by taking advantage of adiabatic deformation of trapping potential, facilitating the design of efficient and stable quantum heat engines.

## Figures and Tables

**Figure 1 entropy-25-00484-f001:**
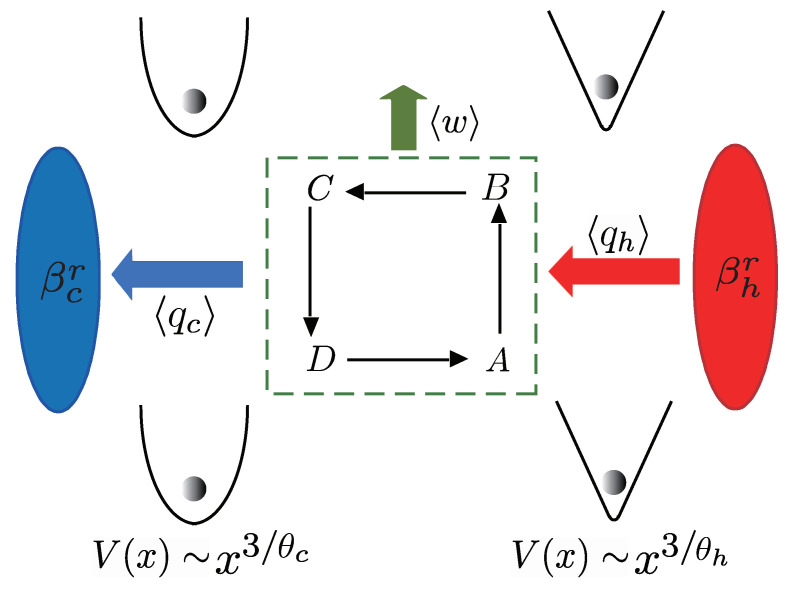
Illustration of a quantum Otto engine cycle with adiabatic deformation of the trap. The engine model works with a single particle confined in the trap. It consists of two isochoric processes A→B and C→D, where the system is weakly coupled to the hot and cold heat reservoirs of constant inverse temperatures $\beta_{h}^{r}$ and $\beta_{c}^{r}\left(>\beta_{h}^{r}\right)$, respectively, and two adiabatic processes B→C and D→A, where the shape of the trap is adiabatically deformed by changing θ in Equation (Equation 1) from $\theta =\theta_{h}$ to $\theta =\theta_{c}$, or vice versa. In each cycle, the average work output (〈w〉) is the difference between the heat absorbed from the hot bath ($\langle q_{h}\rangle$) and the heat released to the cold reservoir ($\langle q_{c}\rangle$); that is, $\langle w\rangle =\langle q_{h}\rangle -\langle q_{c}\rangle$.

**Figure 2 entropy-25-00484-f002:**
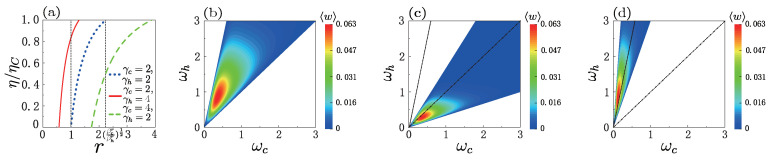
(**a**) Efficiency in unit of $\eta_{C}$ versus ratio *r* (=$\sqrt{\omega_{h}/\omega_{c}})$ for different values of $\gamma_{c,h}$, in which the energy gap is $\omega_{c}=0.36$. The contour maps of 〈w〉 about $\omega_{c}$ and $\omega_{h}$ in the three cases $\gamma_{c}=\gamma_{h}=2,2\gamma_{c}=\gamma_{h}=4$, and $\gamma_{c}=2\gamma_{h}=4$, are, respectively, drawn in (**b**–**d**). The other parameters are $\beta_{c}^{r}=10$ and $\beta_{h}^{r}=2$ in (**a**–**d**). FL The pair values $\left(\gamma_{c},\gamma_{h}\right)=\left(2,4\right)$ [or $\left(\gamma_{c},\gamma_{h}\right)=\left(2,4\right)$] indicate switching from the harmonic potential in the cold isochore to the box trap in the hot isochore (or vice versa), while the values of $\left(\gamma_{h},\gamma_{c}\right)=\left(2,2\right)$ correspond to the case when the shape of the trapping potential is always harmonic along each cycle.

**Figure 3 entropy-25-00484-f003:**
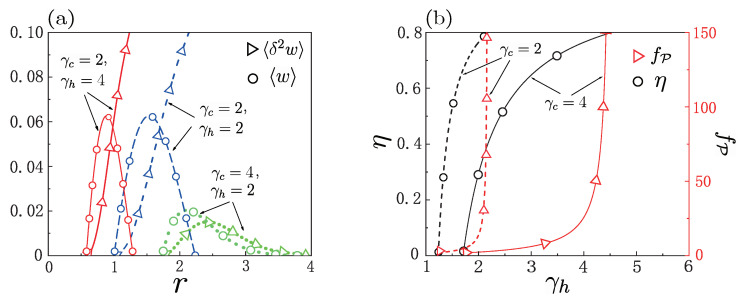
Under quasi-static conditions, work fluctuations $\langle \delta^{2}w\rangle$ and work 〈w〉 versus ratio $r(=\sqrt{\omega_{h}/\omega_{c}})$ for different values of $\gamma_{c,h}$ in (**a**), where the parameter is $\omega_{c}=0.36$. FL in (**a**), the pair values of $\gamma_{c,h}=\left(2,4\right),\left(4,2\right),\left(2,2\right)$ correspond to three respective cases of the potential shapes clarified in Figure 2. The efficiency and relative power fluctuations $f_{P}=\sqrt{\langle \delta^{2}w\rangle /{\langle w\rangle }^{2}}$ versus $\gamma_{h}$ are plotted in (**b**), where the parameters were set to $\omega_{c}=0.2$ and $\omega_{h}=0.85$. The other parameters were $\beta_{c}^{r}=10$ and $\beta_{h}^{r}=2$ in all cases.

**Figure 4 entropy-25-00484-f004:**
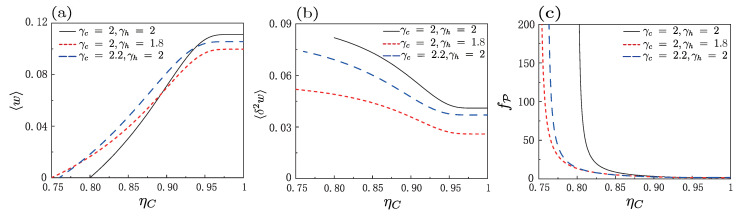
Under quasi-static conditions, (**a**) average work 〈w〉, (**b**) work fluctuations $\langle \delta^{2}w\rangle$, and (**c**) coefficient of variation for power, $f_{P}=\sqrt{\langle \delta^{2}w\rangle /{\langle w\rangle }^{2}}$, as a function of the Carnot efficiency $\eta_{C}$ for different values of $\gamma_{c}$ and $\gamma_{h}$. The parameters were $\omega_{c}=0.12$, $\omega_{h}=0.6$ and $\beta_{h}^{r}=2$.

**Figure 5 entropy-25-00484-f005:**
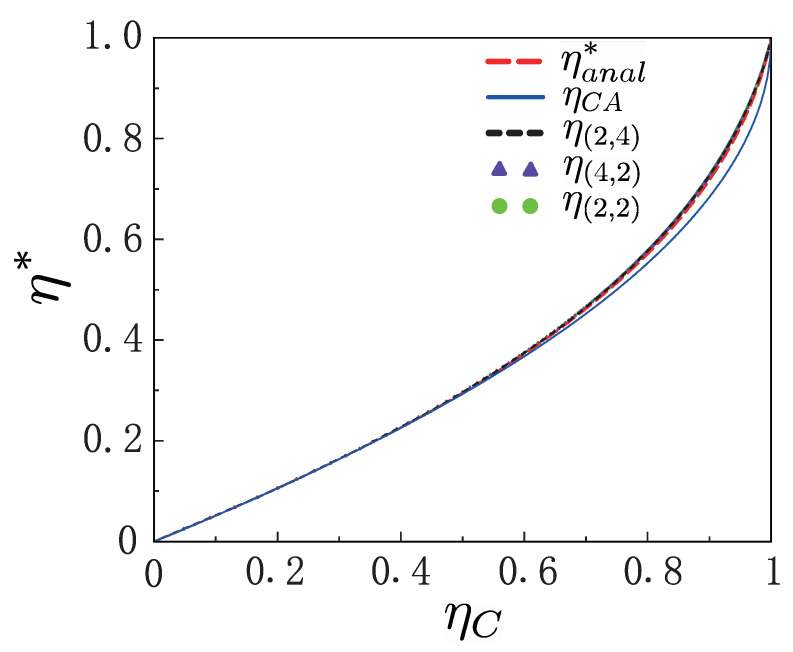
Plots of analytical expression $\eta_{anal}^{*}$ and exact numerical calculations for efficiency at maximum power, and plot of the CA efficiency $\eta_{CA}$. We use $\eta_{\left(\gamma_{c},\gamma_{h}\right)}$ to denote the exact values of optimal efficiency for given $\gamma_{c}$ and $\gamma_{h}$. The inverse temperature of hot bath was $\beta_{h}^{r}=2$.

## Data Availability

The data sets for this study are available upon request from the corresponding author.

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
