# Peer review of "Performance of Quantum Heat Engines Enhanced by Adiabatic Deformation of Trapping Potential"

_entropy, 2023, doi:10.3390/e25030484_

Round 1

Reviewer 1 Report

The authors present a quantum heat engine based on the adiabatic deformation of the trapping potential of a particle. They recover some analytical expressions for the efficiency and the efficiency at maximum-power of the engine as a function of the kind of trapping potential involved. Also, a thorough analysis is made, highlighting pros and cons of the engine, its advantage and disadvantage with respect to other engines.

The work is clearly exposed, and the topic is interesting fro the physics community. Only some minor spells and corrections are needed, as listed here:

-in the abstract the statement "However, such a universality can be reached in the regions beyond the engines without the adiabatic deformation." makes no sense.

-at row 79, in sec.2, the commutator between the annihilation and creation opertors must be corrected (it is [a,a^dag]=1).

-at row 112 substitute "quite long" with "long enough".

-In the title of Sec.4 "Performance".

-At row 205, " is quasi-static limit." must be changed with "is in the quasi-static limit".

-row 341 in App.A, cancel substitute "first law of quantum thermodynamics" with "first law of thermodynamics"

-row 387 cancel "special".

I think that, besides these minor corrections, the work is worth being published on Entropy.

Reviewer 2 Report

     The authors consider a quantum heat engine executing the Otto cycle, where the adiabatic stroke is accomplished by deforming the potential where the working medium is trapped from one shape to another. They show that by appropriately choosing the form of the potential, the fluctuations in the produced work can be decreased, while the efficiency of the engine remains in high levels. This observation thus may lead to the design of more stable quantum heat engines.

     The article is well written and the work is nice, interesting and timely. For these reasons we recommend its publication to Entropy, after the authors consider the following observations:

1.       Regarding Figs. 2 and 3, the authors should clarify the shapes of potential corresponding to the pair values used  (\gamma_c, \gamma_h) = (2, 4), (2, 2), (4, 2).

2.       Regarding Fig. 3(b), the authors can better demonstrate the decrease of fluctuations using some additional examples. For example, for  \gamma_c = 4, they can find a value of \gamma_h < 4 where f_p is halved compared to its value for \gamma_h = 4, while the  efficiency  \eta is (of course) reduced but by a factor smaller than 2 (it actually remains close to the value for \gamma_h = 4). They can also give a corresponding example for \gamma_c = 2.

3.       The authors should rephrase lines 9, 51-52, 283-284, because sincerely we do not understand what exactly they want to say.

4.       The authors may would like to cite the following works, where optimal control is used to minimize the adiabatic stroke time in a harmonic quantum Otto engine:  

(a) D. Stefanatos, Optimal efficiency of a noisy quantum heat engine, Phys. Rev. E 90, 012119 (2014).

(b) D. Stefanatos, Minimum-time transitions between thermal equilibrium states of the quantum parametric oscillator, IEEE Trans. Automat. Control 62, pp. 4290-4297 (2017).

 Some typos:

            Ln 66, \epsilon_n = n\omega

            Formula (5), delete parenthesis is H_D

            Formula (7), in the denominator of the first “big” fraction it is probably \onega_h, not \omega_c.

            Ln. 209, it is probably the square of \tau_cyc in the denominator?

            Ln. 259, “decreased” instead of “deceased”.

Round 2

Reviewer 1 Report

The Authors implemented all the required modifications, I thus suggest the publication of the work in its present form.

Reviewer 2 Report

The authors have incorporated in their text all of our points. One last observation: in the caption of Fig. 2, in the added text, correct the first occurence of the word "isochore".